# DECOUPLING REASONING FROM OBSERVATIONS FOR EFFICIENT AUGMENTED LANGUAGE MODELS

## ABSTRACT

Augmented Language Models (ALMs) blend the reasoning capabilities of Large Language Models (LLMs) with tools that allow for knowledge retrieval and action execution. Existing ALM systems trigger LLM thought processes while pulling observations from these tools in an interleaved fashion. Specifically, an LLM reasons to call an external tool, gets halted to fetch the tool's response, and then decides the next action based on all preceding response tokens. Such a paradigm, though straightforward and easy to implement, often leads to huge computation complexity from redundant prompts and repeated execution. This study addresses such challenges for the first time, proposing a modular paradigm ReWOO (**Re**asoning **WithO**ut **O**bservation) that detaches the reasoning process from external observations, thus significantly reducing token consumption. Comprehensive evaluations across six public NLP benchmarks and a curated dataset reveal consistent performance enhancements with our proposed methodology. Notably, ReWOO achieves $5\times$ token efficiency and $4\%$ accuracy improvement on HotpotQA, a multi-step reasoning benchmark. Furthermore, ReWOO demonstrates robustness under tool-failure scenarios. Besides, decoupling parametric modules from non-parametric tool calls enables instruction fine-tuning to offload LLMs into smaller language models, thus substantially reducing model parameters. Our illustrative work offloads reasoning ability from 175B GPT3.5 into 7B LLaMA successfully, demonstrating the significant potential for truly efficient and scalable ALM systems. Full code, model, and curated data are released for reproduction.[1]

## 1 INTRODUCTION

There is a trending paradigm (Yao et al., 2023; Mialon et al., 2023; Yang et al., 2023; Liang et al., 2023; Qin et al., 2023; Kim et al., 2023; Shinn et al., 2023; Bran et al., 2023) to couple large language models (LLMs) with external plugins or tools, enabling LLMs to interact with environment (Brohan et al., 2023; Nakano et al., 2021) and retrieve up-to-date knowledge. The tool-augmented LLMs, often referred to as augmented language models (ALMs), have fueled applications on autonomous task executions. Existing efforts on ALMs have been widely grounded in the prompting paradigm similar to ReAct (Yao et al., 2023), which interleaves verbal reasoning and tool-calling consecutively.

Such paradigm, however, introduces frequent execution and suspension of LLMs, together with potentially huge cost in terms of token consumption. LLMs generate tokens conditioned on the former context. When interacting with external tools, an LLM has to be halted for tool response. Moreover, the APIs of black-box LLMs, such as ChatGPT, are stateless. To resume the token generation, all the historical tokens (including context prompt, exemplars, all previous reasoning traces and observations) are fed into the LLM, leading to significant prompt redundancy. The commercial LLM service provided by OpenAI charges in terms of token consumption. Thereby, prompt redundancy brings substantial expense to average users. However, to the best of our knowledge, there is no prior work exploring to reduce the token consumption of ALMs.

This paper proposes ReWOO , a novel prompting paradigm for ALMs. As illustrated in Figure 1, ReWOO compartmentalizes the key components of an ALM: step-wise reasoning, tool-calls, and summarization, into three separate modules: Planner, Worker, and Solver. Planner breaks down a

---

[1]Project repo: `https://anonymous.4open.science/r/ReWOO-C1F4`.

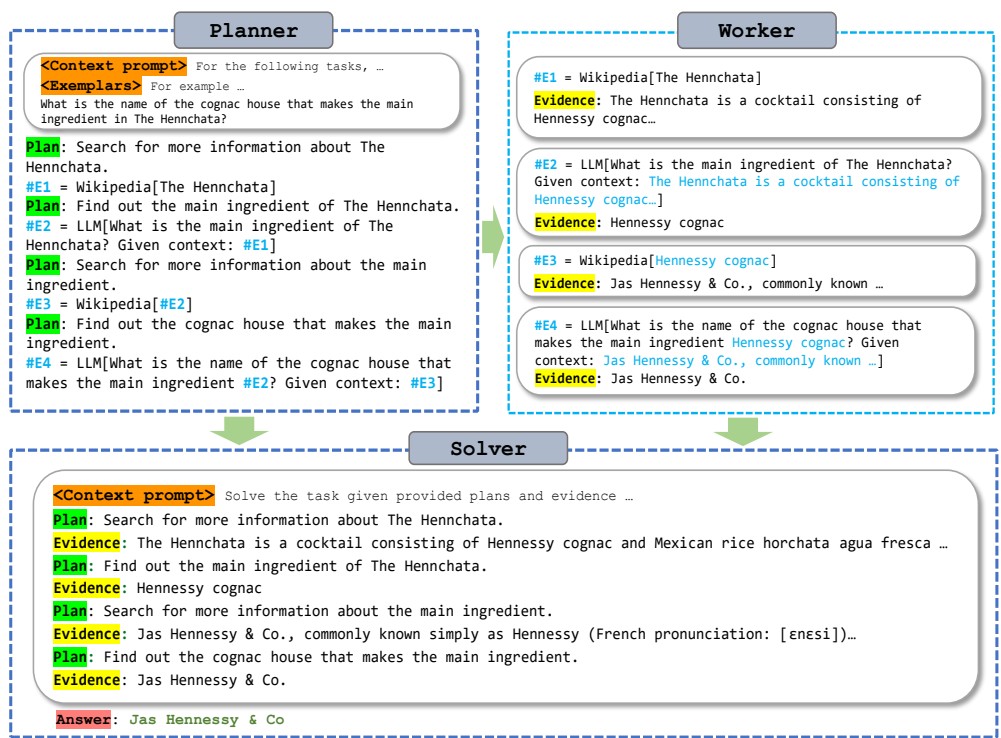

Figure 1: Workflow of ReWOO . Given a question, Planner composes a comprehensive blueprint of interlinked plans prior to tool response. The blueprint instructs Worker to use external tools and collect evidence. Finally, plans and evidence are paired and fed to Solver for the answer.

task and formulates a blueprint of interdependent plans, each of which is allocated to Worker. Worker retrieves external knowledge from tools to provide evidence. Solver synthesizes all the plans and evidence to generate the ultimate answer to the initial task. As shown in Figure 2, ReWOO separates the reasoning process of LLMs from external tools, avoiding the redundancy of interleaved prompts in observation-dependent reasoning, thereby significantly reducing token usage and enhancing ALM system efficiency.

To holistically evaluate ReWOO , we conduct experiments over six multi-step and knowledge-intensive NLP benchmarks and a curated dataset. Evaluation baselines of ReWOO include two non-ALM prompting methods, Direct Prompting, and Chain-of-Thought prompting (CoT) (Wei et al., 2022), and a prevailing ALM paradigm, ReAct (Yao et al., 2023), featuring observation-dependent reasoning. Figure 3 provides an averaged performance over benchmarks in Table 2, demonstrating consistent efficiency gain of ReWOO over its observation-dependent counterpart. Furthermore, we demonstrate the potential of ReWOO for system parameter efficiency through instruction tuning (Taori et al., 2023) and Specialization (Fu et al., 2023). We

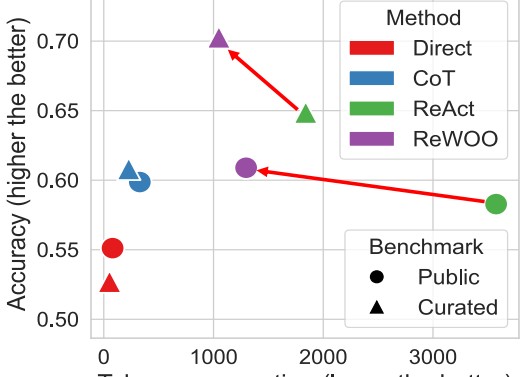

Figure 3: Overall benchmark performance of different methods.

observe that LLaMa 7B fine-tuned with a small number of epochs is on par with GPT3.5 in a zero-shot setup, underscoring the capability of ReWOO to facilitate lightweight and scalable ALM deployment.

**Contributions.** Our contributions to the field of ALM can be summarized as follows: (1) We identify and assess the reasoning ability of LLMs without explicit observations (termed *foreseeable reasoning*). Extensive experiments show that foreseeable reasoning can be harnessed to encourage

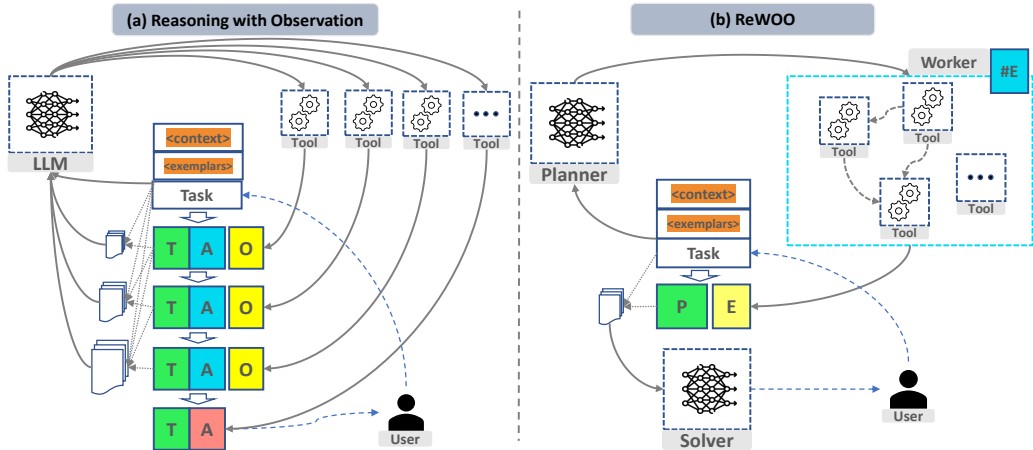

Figure 2: In **(a)** observation-dependent reasoning, the task requested from a user is first wrapped with context prompt and exemplars, then fed into an LLM to initiate a reasoning process. The LLM generates a thought(T) and an action(A), then waits for the observation(O) from tools. The observation is stacked into the prompt history to start the next LLM call. In ReWOO **(b)**, Planner produces at once a list of interdependent plans(P) and calls Worker to fetch evidence(E) from tools. The P and E are combined with the task, and then fed into Solver for the final answer. Note that in (a), the context and exemplars are repeatedly fed into the LLM, resulting in prompt redundancy.

prompt-efficient ALMs. (2) We introduce a modular framework, ReWOO , designed to capitalize on the foreseeable reasoning ability of language models. Comprehensive testing suggests that, compared to the prevalent ALM systems, ReWOO achieves comparable or superior performance while substantially reducing token usage. Moreover, ReWOO exhibits greater robustness in real-world scenarios. (3) We demonstrate a pipeline to offload general foreseeable reasoning from LLMs into smaller language models, enabling the smaller model to utilize unseen tools in zero-shot setups. This research highlights the potential of ReWOO towards scalable and parameter-efficient ALM.

## 2 METHODOLOGY

A salient ability of humans is to predict possible outcomes from to-be-conducted actions. The foreseen outcome of action usually turns out to be instructive enough for adapting and planning on the next steps. Similarly, we design a framework described below.

### 2.1 REWOO WITH PLAN-WORK-SOLVE PARADIGM

**Planner** leverages the foreseeable reasoning of LLMs to compose a solution blueprint. Concretely, it contains consecutive tuples $(Plan, \#E)$ where $Plan$ is a descriptive message of the current step, and $\#E_s$, subscripted by step number $s$, is a special token to store presumably correct evidence from corresponding designated Worker[Instruction]. This paradigm enables ReWOO to tackle multi-step and complex tasks, particularly those where a subsequent step depends on the observations of prior steps, by referring to $\#E_s$ from previous steps in the instructions given to Workers.

**Worker** takes charge of tool-calls. Once Planner provides a blueprint, designated Workers are invoked with instruction input, and populate $\#E_s$ with real evidence or observations.

**Solver** processes all plans and evidence to formulate a solution to the original task or problem, such as providing answers in QA tasks or returning the work status for action requests. We note that prompting Solver to use the provided plans and evidence "with caution" enhances the overall performance of ReWOO . We attribute this improvement to Solver's inherent reasoning ability to resolve simple tasks or partially compensate for failures in the Planner or Worker.

## 2.2 PROMPT REDUNDANCY REDUCTION

ALM systems based on interleaving reasoning and observations suffer undesirable prompt redundancy as depicted in Figure 2 (a). Consider a typical observation-dependent ALM solving a question $Q$ with $k$ reasoning steps to derive the final response $R$. Starting with a context prompt $C$ and a group of $n$ exemplars $\boldsymbol{S} = \{S_i | i \in [1, n]\}$, ALM iteratively generates tuples of Thought, Action, and Observation (TAOs), denoted as $(T_j, A_j, O_j), j \in [1, k]$. Let $\Theta(p)$ denote the number of tokens for a text sequence $p$. The total number of input tokens can be calculated as Eq. (1).

$$\#\texttt{Token}_I^{\texttt{TAO}} = \Theta(C + \boldsymbol{S} + Q) + \sum_{j=1}^{k-1} \Theta\left(C + \boldsymbol{S} + Q + \sum_{t=1}^{j}(T_t + A_t + O_t)\right)$$

$$= \underbrace{k\Theta(Q)}_{\text{Question}} + \underbrace{k\Theta(C)}_{\text{Context}} + \underbrace{k\Theta(\boldsymbol{S})}_{\text{Exemplars}} + \underbrace{\sum_{j=1}^{k-1}(k-j)\Theta(T_j + A_j + O_j)}_{\text{TAOs}} \quad (1)$$

The equation above suggests that duplicated and identical prompts are used as input redundantly. Since $\Theta(C)$ and $\Theta(\boldsymbol{S})$ are usually nontrivial, input tokens quadratically grow oversize as the number of steps $k$ increases, usually leading to token limit excess, ridiculously high computation, and time expenses. On the contrary, ReWOO avoids such interleaving pattern as illustrated in Figure 2 (b). Specifically, let $(P_j, \hat{E}_j, E_j), j \in [1, k]$ be the plan, evidence variable $\#E$ and evidence response at step $j$, The total input tokens for ReWOO is:

$$\#\texttt{Token}_I^{ReWOO} = \Theta(C_{\text{planner}} + \boldsymbol{S} + Q) + \Theta(C_{\text{solver}} + Q + \sum_{j=1}^{k} P_j + E_j)$$

$$\approx \underbrace{2\Theta(Q)}_{Question} + \underbrace{2\Theta(C)}_{\text{Context}} + \underbrace{\Theta(\boldsymbol{S})}_{\text{Exemplars}} + \underbrace{\sum_{j=1}^{k} \Theta(P_j + E_j)}_{\text{PEs}} \quad (2)$$

It is hard to quantitatively measure the difference between the two methods without explicit knowledge of prompting setup. However, if we empirically equalize #TAOs with #PEs, then Eq. (1) differs from Eq. (2) linearly by size of $Q, C, \boldsymbol{S}$ and quadratically by size of $T, A, O$ to the term $k$. As a target task becomes increasingly complicated, introducing more reasoning steps, ReWOO saves substantially larger amounts of computation costs in ALM systems. Note that some LLM-based tools potentially introduce additional token consumption. These tokens are also counted in our experiments.

## 2.3 PARAMETER EFFICIENCY BY SPECIALIZATION

A common concern of ALMs is that binding parametric language models and non-parametric tool calls complicates end-to-end training (Mialon et al., 2023). To mitigate this problem, Toolformer (Schick et al., 2023) fine-tunes language models on tool-augmented corpus in a self-supervised way. Similarly, ReAct makes an attempt to fine-tune reasoning ability on collected reasoning traces from HotpotQA (Yang et al., 2018). These approaches, however, are tested in limited setups. Concretely, Toolformer is limited in an independent sampling of tools, thus failing to function on multi-step reasoning tasks. ReAct's approach in fine-tuning completes thought-action-observations trajectories is unproven to generalize well into unseen tasks or tool set.

ReWOO enables optimization of the generic foreseeable reasoning on a Planner module because no explicit tool response is exposed during fine-tuning. Inspired by recent Specialization framework (Fu et al., 2023), we attempt to elicit foreseeable reasoning from GPT-3.5 and offload into LLaMa 7B (Touvron et al., 2023) as depicted in Figure 4. First, we use text-davinci-003 to infer 4000 (Plan, $\#E$) blueprints on mixed training data of HotpotQA and TriviaQA. Following the bootstrapping method (Zelikman et al., 2022), we sample those leading to correct answers, yielding approximately 2000 Planner instruction data. A pretrained LLaMa 7B is instruction fine-tuned on 52k self-instruct dataset, producing Alpaca (Taori et al., 2023) 7B that approximates general ability of text-davinci-003. Subsequently, we further fine-tune Alpaca-7B on the Planner instruction data to obtain a 7B Planner model specialized in foreseeable reasoning. Finally, we assess the potential of Specialization on multiple benchmarks, replacing the ReWOO Planner with GPT-3.5, Alpaca 7B, and Planner 7B.

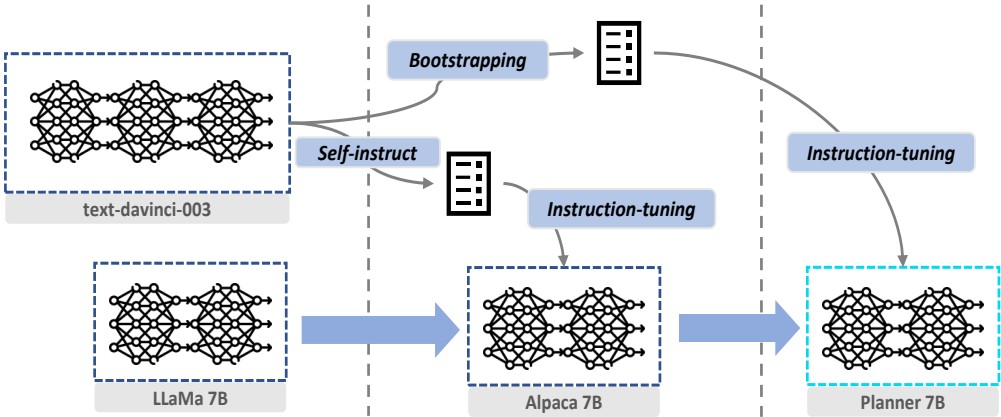

Figure 4: Offloading foreseeable reasoning from GPT-3.5 into Alpaca 7B. A small LLaMa LM is fine-tuned on self-instructed data generated by GPT-3.5, producing Alpaca, endowed with general reasoning ability. Alpaca is then further fine-tuned on blueprints generated by GPT-3.5, leading to Planner 7B, a model specializing in foreseeable reasoning.

## 3 EXPERIMENTS

We evaluate ReWOO against state-of-the-art prompting paradigms across a wide range of NLP benchmarks. To emphasize the necessity of utilizing external tools, we curate a dataset where answering requires up-to-date external knowledge. Notably, ReWOO not only consistently reduces token usage but also matches or even outperforms ReAct in all tasks.

### 3.1 SETUPS

**Tasks and Datasets.** (a) **Common Knowledge and Reasoning.** Such tasks require both domain-specific knowledge and logical reasoning. Four datasets are leveraged for evaluation. *HotpotQA*(Yang et al., 2018), a multi-hop reasoning QA task over diversified domains; *TriviaQA*(Joshi et al., 2017), reading comprehension followed by challenging QAs, where we hide the reading context to encourage searching. *SportsUnderstanding*(Kim, 2022), a factual QA benchmark from BigBench(Ghazal et al., 2013) over in-depth sports domain knowledge; and *StrategyQA*(Geva et al., 2021), an open domain QA task where answers require steps of reasoning. (b) **Arithmetic and Scientific Reasoning.** Such tasks include *GSM8K*(Cobbe et al., 2021) comprised of grade school math problems, and *PhysicsQuestions*(Beatty et al., 2006) on high school physics questions. (c) **Curated.** To challenge ALMs with updated knowledge, we created a QA dataset over State of the Union Address 2023, denoted as SOTUQA. As an instance, "Is Speaker of the House this year older than last year?" expects ALMs to discover Speaker of the House 2023 from provided SOTU document, and 2022 from an online search, then comparing ages. In addition to SOTUQA, we curate a set of tasks aligning with real-world ALM applications (demonstrated in the Appendix), including recommendation for restaurants, stock trading, AI drawing, etc.

**Baselines.** We consider the following prompting paradigms: (a) **Direct Prompt**: a standard zero-shot paradigm that prompts an LLM to directly solve tasks or answer questions. This baseline reflects the language model's ground performance without explicit reasoning or tool utilization. (b) **Chain-of-Thought (CoT)**: prompting an LLM to "think step by step" with an exemplar to demonstrate intermediate verbal reasoning format. This method embodies the models' explicit reasoning ability without tool-calling. (c) **ReAct**: a prevailing prompting paradigm in ALMs as explained in Figure 2. Slightly differing from the original implementation, a short description of the provided tools is appended into the context prompt to enable zero-shot evaluation.

**Exemplars.** For ReWOO Planner, we manually craft $i = \{6, 1, 1\}$ trajectories out of training data from HotpotQA, TriviaQA, and GSM8K, respectively. These exemplars consist of reasoning templates covering information retrieval ("Find out ...", "Search for ..."), comparison ("Compare ... with ... on ..."), equation solving ("Let ... be x, solve ...") and calculating ("Calculate ..."). For

PhysicsQuestions, SportsUnderstanding, and StrategyQA, we shift our interests into systematic generalizability, therefore providing only 1 exemplar from irrelevant benchmarks. The number of reasoning steps $k$ in exemplars is typically 2 or 3. All exemplars used in ReWOO Planner are equivalently provided to ReAct in a thought-action-observation manner. ReAct released the exemplars used on HotpotQA. For a fair comparison, we keep using the same exemplars as ReAct for ReWOO .

**Action Space.** We provide a wide range of tools to assist LLMs in retrieving extra knowledge and interacting with the environment, including (1) **Wikipedia**[query], a search engine for Wikipedia, functioning identically as **search**[entity] in the original ReAct implementation. (2) **Google**[query], search result snippet from Google SERP. (3) **WolframAlpha**[query], search/computation result from Wolfram Alpha API. (4) **LLM**[prompt], a separate single LLM. (5) **Calculator**[prompt], a program-aided LLM (Gao et al., 2022). (6) **SearchDoc**[query], index search over private documents. For curated tasks involving much more diverse and complex real-world interactions, we additionally provide a set of tools like **Location**[query], **Stock**[query], **Twitter**[query], **Yelp**[query], **Email**[request], **TradeStock**[request] and **Draw**[prompt] (See Appendix for examples).

Table 1 shows available tools for different benchmarks. To ensure a fair comparison, we align all available tools provided to ReWOO and ReAct.

**Evaluation Metrics.** Common performance metrics such as exact match (EM) and character-level F1 score are employed in our experiments. Moreover, as observed in (Yao et al., 2023), the correct answers to some benchmark questions are not unique. For example, responding "CA." for the ground-truth "California" should also be considered correct. Therefore, a GPT-4-based scorer is used to measure the semantic accuracy of answers.

| Benchmark | Wiki | LLM | WolfAlf | Calc | Google | SrchDoc |
|-----------|------|-----|---------|------|--------|---------|
| *HotpotQA* | ✓ | ✓ | | | | |
| *TriviaQA* | ✓ | ✓ | | | | |
| *GSM8K* | | ✓ | ✓ | ✓ | | |
| *StrategyQA* | ✓ | ✓ | ✓ | ✓ | ✓ | |
| *PhysicsQA* | ✓ | ✓ | ✓ | ✓ | ✓ | |
| *SportsU.* | ✓ | ✓ | ✓ | ✓ | ✓ | |
| *SOTUQA* | | ✓ | | ✓ | ✓ | ✓ |

Table 1: Available tools for ALMs in different benchmarks.

On the other hand, efficiency can be measured in terms of total token usage in LLMs (including tokens consumed by LLM-based tools), the number of reasoning steps [2] and average token expense in USD for 1k queries.

**Fine-tuning.** We fine-tune 7B LLaMa-based models on a single RTX4090 using LoRA (Hu et al., 2021). Detailed fine-tuninng parameters for Alpaca 7B and Planner 7B are presented in Appendix.

## 3.2 RESULTS AND OBSERVATIONS

### 3.2.1 COMPARISON BETWEEN PROMPTING PARADIGMS

**ReWOO excels over ReAct consistently.** Table 2 shows the main evaluation results on public benchmarks and curated dataset based on gpt-3.5-turbo. Under the ALM setups, we observe the sheer win of ReWOO over ReAct in all benchmarks. Averaging over six public benchmarks, ReWOO is able to reduce token usage by 64% with an absolute accuracy gain of 4.4%. Such results imply the success of ReWOO in eliciting foreseeable reasoning capability of LLMs, as well as the significant efficiency boost of ReWOO against prevailing observation-dependent ALM systems.

**ALMs perform well on curated task** As shown in Table 2(SOTUQA), both ReWOO and ReAct, assisted with external tools, clearly outperform Direct Prompting and CoT. ReWOO outperforms ReAct by 8% absolute accuracy, while consuming 43% less tokens. We believe that the evaluation of document QA like SOTUQA more closely features real-world ALM applications than preceding public NLP benchmarks. In addition, we showcase several ReWOO trajectories in the Appendix, featuring real world ALM applications such as restaurant recommendation and AI painting.

**Extraneous tools compromise ALM performance.** Another finding from Table 2 is that Direct Prompting and CoT, where we don't provide any external tool, outperform both ALM paradigms. This observation leads us to conduct an ablation study on the effect of incrementing tools in ALMs. We start

---

[2] For CoT and ReAct, #steps = #thoughts; For ReWOO , #steps = #plans + 1 including the extra Solver step.

| Dataset | Paradigm | #Tools | $n$ | Acc | F1 | EM | #Tokens | #Steps | $Cost_{1k}$ |
|---------|----------|--------|-----|-----|-----|-----|---------|--------|-------------|
| *HotpotQA* 1000 | Direct | 0 | 0 | 37.8 | 36.2 | 28.0 | 55.5 | 1.00 | 0.11 |
| | CoT | 0 | 1 | 41.6 | 30.8 | 22.4 | 481.9 | 1.79 | 0.96 |
| | REACT | 2 | 6 | 40.8 | 39.6 | **32.2** | 9795.1 | 4.97 | 19.59 |
| | ReWOO | 2 | 6 | **42.4** | **40.1** | 30.4 | 1986.2 | 4.45 | 3.97 |
| *TriviaQA* 1000 | Direct | 0 | 0 | 80.6 | 74.0 | 64.2 | 43.4 | 1.00 | 0.09 |
| | CoT | 0 | 1 | 78.6 | 71.7 | 60.1 | 199.2 | 2.08 | 0.40 |
| | REACT | 2 | 1 | 59.4 | 53.2 | 47.4 | 4212.9 | 5.21 | 8.43 |
| | ReWOO | 2 | 1 | **66.6** | **60.6** | **51.8** | 1340.9 | 3.55 | 2.68 |
| *GSM8K* 1000 | Direct | 0 | 0 | 26.8 | 14.4 | — | 101.1 | 1.00 | 0.20 |
| | CoT | 0 | 1 | 67.4 | 62.7 | — | 495.6 | 3.45 | 0.99 |
| | REACT | 3 | 1 | 62.0 | **37.3** | — | 1874.3 | 2.86 | 3.75 |
| | ReWOO | 3 | 1 | **62.4** | 36.2 | — | 1089.3 | 3.21 | 2.18 |
| *StrategyQA* 300 | Direct | 0 | 0 | 64.6 | 64.6 | 64.6 | 41.8 | 1.00 | 0.08 |
| | CoT | 0 | $1^\dagger$ | 56.0 | 56.0 | 56.0 | 170.5 | 1.85 | 0.34 |
| | REACT | 5 | $1^\dagger$ | 64.6 | 64.6 | 64.6 | 1686.3 | 2.58 | 3.37 |
| | ReWOO | 5 | $1^\dagger$ | **66.6** | **66.6** | **66.6** | 1287.1 | 3.20 | 2.57 |
| *PhysicsQA* 53 | Direct | 0 | 0 | 52.8 | 12.6 | — | 132.2 | 1.00 | 0.26 |
| | CoT | 0 | $1^\dagger$ | 62.2 | 15.2 | — | 346.8 | 3.07 | 0.69 |
| | REACT | 5 | $1^\dagger$ | 64.1 | **16.2** | — | 2163.3 | 2.77 | 4.33 |
| | ReWOO | 5 | $1^\dagger$ | **66.0** | 14.0 | — | 1225.7 | 2.56 | 2.45 |
| *SportsU.* 300 | Direct | 0 | 0 | 68.0 | 68.0 | 68.0 | 47.63 | 1.00 | 0.10 |
| | CoT | 0 | $1^\dagger$ | 53.3 | 47.5 | 45.3 | 215.9 | 1.78 | 0.43 |
| | REACT | 5 | $1^\dagger$ | 58.6 | 51.9 | 49.3 | 1720.0 | 2.64 | 3.44 |
| | ReWOO | 5 | $1^\dagger$ | **61.3** | **55.8** | **55.3** | 854.2 | 3.04 | 1.71 |
| *SOTUQA* Curated | Direct | 0 | 0 | 52.7 | 15.3 | — | 52.2 | 1.00 | 0.10 |
| | CoT | 0 | $1^\dagger$ | 60.8 | 21.2 | — | 227.4 | 2.08 | 0.45 |
| | REACT | 5 | $1^\dagger$ | 64.8 | 42.7 | — | 1840.3 | 2.43 | 3.68 |
| | ReWOO | 5 | $1^\dagger$ | **70.2** | **44.8** | — | 1048.8 | 2.24 | 2.09 |

Table 2: Evaluation results on public NLP benchmarks. For HotpotQA, TriviaQA, and GSM8K, prompts are configured with tools and exemplars labeled from the same benchmarks; Other tasks align with practical scenarios, where we use a static out-of-task exemplar to instruct output format (which can be seen as zero-shot), and a common large tool set. $n$ denotes the number of exemplars. †: Out-of-task exemplar. Underline: Best performing paradigm. **Bold**: Best performing ALM.

with the same setups for HotpotQA while incrementally adding one extra tool to ReWOO and ReAct. Figure 5 shows that while a powerful tool like Google temporarily boosts accuracy, the general trend goes down as we introduce more tools in-context. Qualitatively, we investigate 20 questions where 2-tool ReWOO succeeds while 7-tool ReWOO fails, observing that 17 of the trajectories involve tool misuse, such as employing **Yelp**[query] to search for a celebrity. This experiment indicates that unnecessary tools are harmful to ALMs by potentially introducing erroneous or extraneous contents.

**ReWOO is relatively robust upon tool failure.** Suboptimal tool responses are common in ALM systems. To compare the robustness of ReWOO and ReAct, we overwrite all tool responses with "No evidence found.". Table 3 implies that ReAct-like ALM systems are highly fragile when intermediate tools fail. On the other hand, ReWOO is less compromised under tool failures at a smaller cost.

**Conversation-aligning RLHF in ALM.** To explore the effect of RLHF, we replace gpt-3.5-turbo based LLMs used in HotpotQA with text-davinci-003. Table 3 shows that text-davinci-003 outperforms gpt-3.5-turbo with fewer steps and token usage, implying that conversation RLHF slightly harms commonsense reasoning ability of ALMs.

| Method | Acc | #Tokens | $Cost_{1k}$ |
|--------|-----|---------|-------------|
| *Normal* | | | |
| REACT | 40.8 | 9795.1 | 21.29 |
| ReWOO | **42.4** | 1986.2 | 3.97 |
| $\Delta$ *with tool-failure* | | | |
| REACT | -40.8 | +851.1 | +1.70 |
| ReWOO | **-29.2** | -110.8 | -0.22 |
| $\Delta$ *with gpt-3.5-turbo $\rightarrow$ text-davinci-003* | | | |
| REACT | +1.7 | -466.8 | -0.93 |
| ReWOO | **+2.6** | -90.5 | -0.18 |

Table 3: Performance change on HotpotQA when (1) all tools return "No evidence found" (2) replacing LLM.

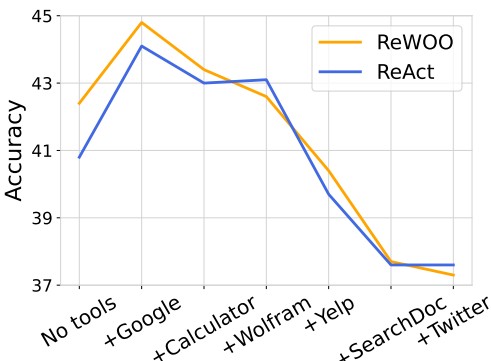

Figure 5: Performance degraded on HotpotQA when incrementally adding extraneous tools.

### 3.3 FINE-TUNING AND SPECIALIZATION OF LLM

Following the Specialization framework from Figure 4, we obtain Alpaca 7B and Planner 7B approximating general ability and foreseeable reasoning from GPT3.5, respectively. Both LMs are compared against the original GPT-3.5 performance in a zero-shot setup. Figure 6 reflects that these methods, when plugged into the Planner module, are on par with $25\times$ larger GPT-3.5 in HotpotQA, TriviaQA and StrategyQA benchmarks. Furthermore, general accuracy gain from Alpaca 7B to Planner 7B implies effectiveness of Specialization. Qualitatively, although demonstrated tools in training instructions only involve **Wikipedia**[query] and **LLM**[prompt], we surprisingly observe that, if paired with in-context descriptions, Planner 7B is increasingly capable

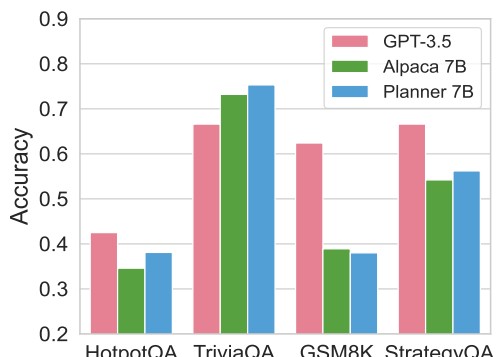

Figure 6: Performance alternating LLMs in Re-WOO . Planner 7B is tuned on HotpotQA and TriviaQA

of reasoning with **Google**[query] and **Calculator**[prompt] than Alpaca. Further efforts are required to push the limits of Specialization, which we leave for future studies. Most importantly, our results illustrate the potential of ReWOO paradigm in offloading general foreseeable reasoning into distilled small language models, substantially improving system parameter efficiency and scalability.

## 4 LIMITATIONS AND FUTURE WORK

We notice that for certain tasks under unusual contexts, fully relying on foreseeable reasoning becomes impractical. Consider the following task from AlfWorld(Shridhar et al., 2020):

> *You are in the middle of a room. Looking quickly around you, you see a drawer 2, a shelf 5, a drawer 1, a shelf 4, a sidetable 1, a drawer 5, a shelf 6, a shelf 1, a shelf 9, a cabinet 2, a sofa 1, a cabinet 1, a shelf 3, a cabinet 3, a drawer 3, a shelf 11, a shelf 2, a shelf 10, a dresser 1, a shelf 12, a garbagecan 1, a armchair 1, a cabinet 4, a shelf 7, a shelf 8, a safe 1, and a drawer 4. Your task is to: put some vase in safe.*

Since a Planner has no prior knowledge about the environment, it has to enumerate all possible plans that can potentially lead to *some vase*.

The example above implies that a robust ALM system should not be built on a singleton – it looks promising to wire different nodes of LLMs, tools, and sub-models into a directed acyclic graph (DAG) so that each node functions for its predesignated tasks organically. [3] Directions to further improve

---

[3]Recent projects like LangChain(lan, 2023) have, to some extent, featured this idea.

the efficiency and performance of such ALM systems include (1) **Offloading specialized abilities from foundation LLMs into smaller models.** Section 3.3 demonstrate the possibility for small LMs specializing Fu et al. (2023) in general foreseeable reasoning. We expect that with a greater number of open domain instructions, foreseeable reasoning can be even more holistically offloaded. Other parametric nodes in the DAG, such as a Solver, can be fine-tuned alike. (2) **Tool representation learning.** In many cases from HotpotQA, Wikipedia and Google can both lead to the correct answer, indicating a certain level of similarity between those tools. We can set up a model to minimize the energy among similar-functioning Workers. Tool representations allow us to parametrize the whole ALM system and therefore enabling end-to-end training. (3) **Graph optimization.** Furthermore, we should be able to optimize DAG execution through multiple graph and concurrency algorithms.

## 5 RELATED WORK

**Tool-augmented LLMs.** When prompted properly, LLMs demonstrate the ability of reasoning to solve tasks using evidence and logic, such as commonsense reasoning, mathematical reasoning, and symbolic reasoning (Mialon et al., 2023). Several works injects the intermediate reasoning steps with diverse tools, enabling LLMs to retrieve up-to-date world knowledge and solving more complex tasks. Search APIs are leveraged to avoid hallucinations and provide comprehensive information for more trustworthy text generation (Yao et al., 2023; Nakano et al., 2021; Lazaridou et al., 2022). High-level robotics APIs are used to instruct robotics to finish physical world tasks (Brohan et al., 2023; Liang et al., 2022; Driess et al., 2023; Vemprala et al., 2023). Calculator (Cobbe et al., 2021), code interpreter (Gao et al., 2022), and mathematical prover (Jiang et al., 2022) are used to fix the calculation error, execute the generated code, and prove the complex mathematical theory, respectively. There are also works that use multiple tools to solve various natural language processing and computer vision tasks, such as Toolformer (Schick et al., 2023) and Visual ChatGPT (Wu et al., 2023). In addition, the task can be decomposed, and the problem can be better solved using multi-step reasoning and actions, such as ReAct (Yao et al., 2023), ART (Paranjape et al., 2023), MM-ReAct (Yang et al., 2023), and TaskMatrix.AI (Liang et al., 2023). Our work poses a new perspective to tool-augmented LLMs for large-scale real-world applications.

**Efficient LLMs.** Efficient LLMs is a lasting research topic, particularly with the prevailing of ChatGPT. Various approaches(Hu et al., 2021; Liu et al., 2022; Peng et al., 2023; Taori et al., 2023; Lester et al., 2021; Wingate et al., 2022; Houlsby et al., 2019; He et al., 2021; Cheng et al., 2023) have been proposed to reduce the cost of fine-tuning and deploying LLMs. A prevailing direction is to reduce model scale, e.g., using instruction tuning(Peng et al., 2023; Taori et al., 2023) to align a small and locally-hosted LLM with the assistance from large-scale black-box LLMs. The computation cost during tuning can be further reduced by LoRA(Hu et al., 2021), adapters(Houlsby et al., 2019; He et al., 2021), prompt tuning(Wingate et al., 2022; Lester et al., 2021), etc. However, these approaches often involve nontrivial modifications of model structures, hindering the application to black-box LLMs. In contrast, prompt engineering for efficient LLMs, though rarely studied, is flexible and straightforward. It demands no internal information of LLMs and can be readily applied to any off-the-shelf black-box language models like OpenAI ChatGPT and Google PaLM. Following this direction, our work gives the first exploration of prompting for efficient tool-augmented LLMs.

## 6 CONCLUSION

We present ReWOO , a modular ALM framework to solve multi-step reasoning tasks efficiently by decoupling reasoning from tool feedback and observations. Theoretical decomposition of prompt tokens establishes that ReWOO is able to substantially reduce prompting redundancy in prevailing Thought-Action-Observation ALM systems. Comprehensive experiments on both public NLP benchmarks and curated tasks reveal superior performance of ReWOO in achieving boosted performance with much less token consumption. A side study also shows that ReWOO has relatively robust performance under tool-failure cases. Our study further unveils the potential for generic reasoning offloading via instruction tuning and specialization. Future improvements beyond ReWOO based ALM systems involve modular LLM fine-tuning, tool representation learning, and system graph learning and optimization. We demonstrate that our work lays a solid foundation for these advancements, inching us closer to truly scalable AGI.

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

## A  STATEMENT OF BROADER IMPACT

The broader impacts of our proposed `ReWOO` methodology and entailed ALM systems extend to several fields and real-world scenarios. Firstly, the efficiency gains enabled by `ReWOO`'s modular paradigm not only conserve computational resources but also pave the way for more extensive and diverse applications of ALMs, even on systems with limited computational capabilities. An ALM built with similar paradigms could significantly expand the opportunities for businesses, researchers, and developers in resource-constrained environments. For upper-stream LLM providers like OpenAI, widely adopting `ReWOO` in their plugin systems could benefit in incremental computation efficiency and system throughput.

Secondly, our methodology's robustness under tool-failure scenarios is a significant step forward in ALM technology's reliability, contributing to more resilient AI systems capable of managing unseen and unpredictable real-world situations. This resilience could translate into a more reliable service for users, enhancing user experience and trust.

Thirdly, our offloading and specialization pipeline exemplifies model parameter reduction in ALM systems, benefiting deployment and scalability. This provides a route to more sustainable AI applications, spreading locally accessible ALMs into a wider range of research facilities and industries.

Potential risks of `ReWOO` generally align with that of a lightweight AI system. The increase in ALM accessibility and efficiency might lead to the misuse of these models, such as creating harmful or biased content. Furthermore, the specialization process into smaller LLMs comes at the cost to compromise other capabilities. We urge to prevent users from placing undue trust in such offloaded smaller LLMs without fully understanding such trade-off.

## B  ADDITIONAL OBSERVATIONS

### B.1  TOKEN DECOMPOSITION

We decompose the token usage of different prompt paradigms on HotpotQA into different components – context prompts, exemplars, and intermediate steps. Figure B.1 shows that, compared to `ReWOO`, ReAct consumes significantly more tokens in exemplars. We attribute this gap to the following reasons: (1) Exemplars are repetitively prompted for the number of reasoning steps (approximately 5 times in the HotpotQA experiment), whereas `ReWOO` has no such repetition; (2) Exemplars used in the ReAct paradigm inevitably include *Observation* at each reasoning step, which can occasionally be a lengthy result from a Wikipedia page. In contrast, exemplars used in `ReWOO` don't contain any explicit observations.

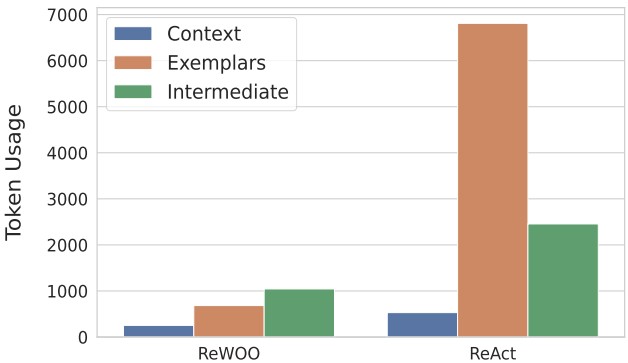

Figure 7: Decomposition of token usage on HotpotQA.

### B.2  HOW DOES ReWOO OUTPERFORM ReAct?

The superior accuracy of `ReWOO` over ReAct on multiple benchmarks elicits surprise, given our human propensity to base future actions on current and previous observations for more accurate moves, rather than formulating comprehensive plans in advance. To further explore such seemingly

paradoxical findings, we randomly select 100 failure cases from HotpotQA for both methods and probe into the failure reasons. We label each trajectory with one or more of the following tags: (1) Bad Reasoning, where we find the reasoning trajectory misleads or deviates from the question; (2) Tool Inefficacy, observed when Wikipedia[query] is unable to retrieve pertinent information. (3) Token Excess, wherein the maximum 4096 token limit of GPT-3.5 is reached, typically as a result of an excessive number of reasoning steps; (4) Answer Miss, where all reasoning and tool responses are considered right, but the model fails to deduce the correct answer; (5) Ambiguous Question, a scenario in which the original task either contains an erroneous or outdated ground-truth label or accepts multiple valid answers.

|  | Bad Reason. | Tool Inefficacy | Token Excess | Answer Miss | Ambiguous Q. |
|---|---|---|---|---|---|
| **ReAct** | 76 | 20 | 18 | 3 | 17 |
| **ReWOO** | 51 | 29 | 0 | 11 | 17 |

We find that a bad tool response easily ruins the reasoning trace of ReAct, resulting in an infinite action loop or repetition. In fact, we observe a common scenario in ReAct when toolA fails, and ReAct attempts to invoke toolB. Then if toolB fails as well, ReAct turns back to invoke toolA again, and so on until hitting the token limit. Besides, we find that when the number of reasoning steps goes above four, the context prompt of ReAct becomes extremely lengthy, sometimes leading to deviation from the original problem.

On the other hand, `ReWOO` can usually generate a reasonable planning trajectory independent from tool failures. Such plans, though "reasonable", can sometimes be ineffective because of incorrect expectations. For instance, `ReWOO` uses Wikipedia[query] to retrieve information about a person, then it assumes that the age of that person exists in the search results, thereby using an LLM[prompt] later to extract his age. Such an assumption can be erroneous when the Wikipedia results don't actually contain his age. Besides, we observe that Solver sometimes gives the wrong conclusion even if all plans and evidence are solid. We think a better Solver prompt or providing a simple exemplar could mitigate this issue.

## C  IMPLEMENTATION DETAILS

### C.1  INSTRUCTION TUNING

When instruction-tuning Alpaca 7B, we use the low-rank adaptation (LoRA) following a framework implemented in `https://github.com/tloen/alpaca-lora`. This trick allows us to fine-tune and specialize Planner 7B model on a single RTX 4090. We use batch-size=128, learning rate=1e-4, cutff_len=1024 with lora_r=8, and train upon Alpaca 7B (`https://huggingface.co/tloen/alpaca-lora-7b`) for 10 epochs. The specialized Planner 7B is uploaded to `https://huggingface.co/rewoo/planner_7B`. This model can further benefit from more instruction planning data and deliberate training setups.

### C.2  PROMPTS

Hereby we disclose the context prompts and exemplars used in `ReWOO`. The tool descriptions and exemplars are subject to setups at run time. Notably, `ReWOO` is a general paradigm and prompts are not necessarily fixed. We encourage readers and users to adjust the prompts tailored to their own needs.

---

**–PLANNER–**

For the following task, make plans that can solve the problem step by step. For each plan, indicate which external tool together with tool input to retrieve evidence. You can store the evidence into a variable #E that can be called by later tools. (Plan, #E1, Plan, #E2, Plan, ...)

Tools can be one of the following:

(1) *Google[input]*: Worker that searches results from Google. Useful when you need to find short and succinct answers about a specific topic. The input should be a search query.

(2) *Wikipedia[input]*: Worker that search for similar page contents from Wikipedia. Useful when you need to get holistic knowledge about people, places, companies, historical events, or other subjects. The response is long and might contain some irrelevant information. The input should be a search query.

(3) *WolframAlpha[input]*: Useful when you need to solve a Mathematical or Algebraic equation. Input should be an equation or function.

(4) *Calculator[input]*: A calculator that can compute arithmetic expressions. Useful when you need to perform math calculations. Input should be a mathematical expression

(5) *LLM[input]*: A pretrained LLM like yourself. Useful when you need to act with general world knowledge and common sense. Prioritize it when you are confident in solving the problem yourself. Input can be any instruction.

(6) *SearchSOTU[input]*: A vector store that searches for similar and related content in a document: state_of_the_union. The result is a huge chunk of text related to your search but can also contain irrelevant info. The input should be a search query.

For example,

Task: Thomas, Toby, and Rebecca worked a total of 157 hours in one week. Thomas worked x hours. Toby worked 10 hours less than twice what Thomas worked, and Rebecca worked 8 hours less than Toby. How many hours did Rebecca work?

Plan: Given Thomas worked x hours, translate the problem into algebraic expressions and solve with Wolfram Alpha. #E1 = WolframAlpha[Solve $x + (2x - 10) + ((2x - 10) - 8) = 157$]

Plan: Find out the number of hours Thomas worked. #E2 = LLM[What is $x$, given #E1]

Plan: Calculate the number of hours Rebecca worked. #E3 = Calculator[$(2 * $ #E2 $- 10) - 8$]

Begin! Describe your plans with rich details. Each Plan should be followed by only one #E.

**\<Your task\>**

---

**− SOLVER −**

Solve the following task or problem. To solve the problem, we have made step-by-step Plan and retrieved corresponding Evidence to each Plan. Use them with caution since long evidence might contain irrelevant information.

**\<Plans + Evidence\>**

Now solve the question or task according to provided Evidence above. Respond with the answer directly with no extra words.

**\<Your task\>**

---

## D    TRAJECTORIES

In the following section, we show several reasoning trajectories of ReAct and `ReWOO` on the evaluated datasets. All the prompting and tools are set up as discussed in Section 3.1.

### D.1    HOTPOTQA

*ReAct*

---

Question: Who made the 1989 comic book, the film version of which Jon Raymond Polito appeared in?

---

 Thought : I need to search Jon Raymond Polito, find the film version he appeared in, then find who made the 1989 comic book.

 Action : Search[Jon Raymond Polito]

 Observation : Jon Raymond Polito (December 29, 1950 – September 1, 2016) was an American character actor. In a film and television career spanning 35 years, he amassed over 220 credits. Notable television roles included Detective Steve Crosetti in the first two seasons of Homicide: Life on the Street and as Phil Bartoli on the first season of Crime Story. He also appeared in several films including The Rocketeer, The Crow and Gangster Squad, as well as his work with the Coen brothers. He appeared in five of their films, including Miller's Crossing, Barton Fink and The Big Lebowski. Polito also portrayed legendary "hungry i" nightclub impresario Enrico Banducci in a large supporting role in Tim Burton's 2014 film Big Eyes starring Amy Adams and Christoph Waltz.

 Thought : Jon Raymond Polito appeared in The Rocketeer, The Crow, Gangster Squad, Miller's Crossing, Barton Fink, The Big Lebowski, and Big Eyes. I need to search The Rocketeer and find who made the 1989 comic book.

 Action : Search[The Rocketeer]

Observation : The Rocketeer (released internationally as The Adventures of the Rocketeer) is a 1991 American period superhero film from Walt Disney Pictures and Touchstone Pictures. It was produced by Charles Gordon, Lawrence Gordon, and Lloyd Levin, directed by Joe Johnston, and stars Billy Campbell, Jennifer Connelly, Alan Arkin, Timothy Dalton, Paul Sorvino, and Tiny Ron Taylor. It is based on the character of the same name created by comic book artist and writer Dave Stevens. Set in 1938 Los Angeles, California, The Rocketeer tells the story of stunt pilot, Cliff Secord, who discovers a hidden rocket pack that he thereafter uses to fly without the need of an aircraft. His heroic deeds soon attract the attention of Howard Hughes and the FBI, who are hunting for the missing rocket pack, as well as the Nazi operatives that stole it from Hughes. Development for The Rocketeer started as far back as 1983, when Stevens sold the film rights to the character. Steve Miner and William Dear considered directing The Rocketeer before Johnston signed on. Screenwriters Danny Bilson and Paul De Meo had creative differences with Disney, causing the film to languish in development hell. The studio also intended to change the trademark helmet design; Disney CEO Michael Eisner wanted a straight NASA-type helmet, but Johnston convinced the studio otherwise. Johnston also had to convince Disney to let him cast actor Billy Campbell in the lead role. Filming for The Rocketeer lasted from September 19, 1990, to January 22, 1991. The visual effects sequences were created and designed by Industrial Light & Magic and supervised by animation director Wes Takahashi. The film was released on June 21, 1991, and received positive reviews from critics. Plans for Rocketeer sequels were abandoned after the film underperformed at the box office, grossing only $46 million on a $35$ million budget. However, a television series based on the film, with Campbell reprising his role, premiered on Disney Junior in November 2019. In early 1991, Toy Biz had the rights to a massive Rocketeer toy line, including different versions of Cliff; this was akin to the Kenner Products Batman toy lines. Several prototypes were made, but the toy line was promptly shut down; Disney pulled the license from Toy Biz leaving only Just Toys having the rights.

Thought : The Rocketeer is based on the character of the same name created by comic book artist and writer Dave Stevens. So Dave Stevens made the 1989 comic book.

Action : Finish[ Dave Stevens ]

*ReWOO*

Question: Who made the 1989 comic book, the film version of which Jon Raymond Polito appeared in?

**−PLANNER−**

Plan : Search for more information about Jon Raymond Polito.

#E1 = Wikipedia[Jon Raymond Polito]

Plan : Search for more information about the 1989 comic book.

#E2 = LLM[What is the name of the 1989 comic book? Given context: #E1 ]

Plan : Search for more information about the film version of the comic book.

#E3 = Wikipedia[ #E2 film version]

Plan : Find out who made the 1989 comic book.

#E4 = LLM[Who made the 1989 comic book #E2 ? Given context: #E3 ]

**−SOLVER−**

Plan : Search for more information about Jon Raymond Polito.

Evidence : Jon Raymond Polito (December 29, 1950 – September 1, 2016) was an American character actor. In a film and television career spanning 35 years, he amassed over 220 credits. Notable television roles included Detective Steve Crosetti in the first two seasons of Homicide: Life on the Street and as Phil Bartoli on the first season of Crime Story. He also appeared in several films including The Rocketeer, The Crow and Gangster Squad, as well as his work with the Coen brothers. He appeared in five of their films, including Miller's Crossing, Barton Fink and The Big Lebowski. Polito also portrayed legendary "hungry i" nightclub impresario Enrico Banducci in a large supporting role in Tim Burton's 2014 film Big Eyes starring Amy Adams and Christoph Waltz.

Plan : Search for more information about the 1989 comic book.

Evidence : The Rocketeer.

Plan : Search for more information about the film version of the comic book.

Evidence : The Rocketeer (released internationally as The Adventures of the Rocketeer) is a 1991 American period superhero film from Walt Disney Pictures and Touchstone Pictures. It was produced by Charles Gordon, Lawrence Gordon, and Lloyd Levin, directed by Joe Johnston, and stars Billy Campbell, Jennifer Connelly, Alan Arkin, Timothy Dalton, Paul Sorvino, and Tiny Ron Taylor. It is based on the character of the same name created by comic book artist and writer Dave Stevens. Set in 1938 Los Angeles, California, The Rocketeer tells the story of stunt pilot, Cliff Secord, who discovers a hidden rocket pack that he thereafter uses to fly without the need of an aircraft. His heroic deeds soon attract the attention of Howard Hughes and the FBI, who are hunting for the missing rocket pack, as well as the Nazi operatives that stole it from Hughes. Development for The Rocketeer started as far back as 1983, when Stevens sold the film rights to the character. Steve Miner and William Dear considered directing The Rocketeer before Johnston signed on. Screenwriters Danny Bilson and Paul De Meo had creative differences with Disney, causing the film to languish in development hell. The studio also intended to change the trademark helmet design; Disney CEO Michael Eisner wanted a straight NASA-type helmet, but Johnston convinced the studio otherwise. Johnston also had to convince Disney to let him cast actor Billy Campbell in the lead role. Filming for The Rocketeer lasted from September 19, 1990, to January 22, 1991. The visual effects sequences were created and designed by Industrial Light & Magic and supervised by animation director Wes Takahashi. The film was released on June 21, 1991, and received positive reviews from critics. Plans for Rocketeer sequels were abandoned after the film underperformed at the box office, grossing only $46 million on a $35$ million budget. However, a television series based on the film, with Campbell reprising his role, premiered on Disney Junior in November 2019. In early 1991, Toy Biz had the rights to a massive Rocketeer toy line, including different versions of Cliff; this was akin to the Kenner Products Batman toy lines. Several prototypes were made, but the toy line was promptly shut down; Disney pulled the license from Toy Biz leaving only Just Toys having the rights.

Plan : Find out who made the 1989 comic book.

Evidence : Dave Stevens.

Dave Stevens.

## D.2 TRIVIAQA

***ReAct***

Question: Who featured on Melanie C's number one single Never Be The Same Again in 2000, and died in a car accident in 2002?

Thought : I need to search Melanie C's number one single Never Be The Same Again, find who featured on it, then find if they died in a car accident in 2002.

Action : Search[Never Be The Same Again]

Observation : "Never Be the Same Again" is a song by British singer-songwriter Melanie C from the British girl group Spice Girls, featuring American rapper Lisa "Left Eye" Lopes of American girl group TLC. It was released on 20 March 2000 as the third single from her first solo album, Northern Star (1999). The song was co-written by Melanie C, producer Rhett Lawrence, Paul F. Cruz and Lopes. "Never Be the Same Again" entered at the top of the UK Singles Chart, beating Moloko's "The Time Is Now" to the summit and it was Melanie C's first solo single to reach number one. It sold 144,936 copies in its first week and was Britain's eighteenth best-selling song of 2000. The song was successful in other markets, topping the charts in seven countries and receiving positive acclaim. By April 2021, the song had sold more than 477,000 copies in UK. The music video was directed by Francis Lawrence. It shows Melanie C waking up in a futuristic home and practicing tai chi with Lopes.

Thought : Lisa "Left Eye" Lopes featured on Melanie C's number one single Never Be The Same Again in 2000. I need to search Lisa "Left Eye" Lopes and find if she died in a car accident in 2002.

Action : Search[Lisa "Left Eye" Lopes]

Observation : Lisa Nicole Lopes (May 27, 1971 – April 25, 2002), better known by her stage name Left Eye, was an American rapper and singer. She was a member of the R&B girl group TLC, alongside Tionne "T-Boz" Watkins and Rozonda "Chilli" Thomas. Besides rapping and singing backing vocals on TLC recordings, Lopes was the creative force behind the group, receiving more co-writing credits than the other members. She also designed some of their outfits and the stage for 'Fan Mail Tour' and contributed to the group's image, album titles, artworks, and music videos. Through her work with TLC, Lopes won four Grammy Awards.During her brief solo career, Lopes scored two US top 10 singles with "Not Tonight" and "U Know What's Up", as well as one UK number-one single with "Never Be the Same Again", the latter a collaboration with Melanie C of the British girl group Spice Girls. She also produced another girl group, Blaque, who scored a platinum album and two US top 10 hits. Lopes remains the only member of TLC to have released a solo album. On April 25, 2002, Lopes was killed in a car crash while shooting a documentary in Honduras. The vehicle she was in swerved off the road to avoid hitting another vehicle that stopped abruptly in front of them. The documentary was released as The Last Days of Left Eye and aired on VH1 in May 2007.

Thought : Lisa "Left Eye" Lopes featured on Melanie C's number one single Never Be The Same Again in 2000 and died in a car accident in 2002.

Action : Finish[ Lisa "Left Eye" Lopes ]

*ReWOO*

Question: Who featured on Melanie C's number one single Never Be The Same Again in 2000, and died in a car accident in 2002?

**–PLANNER–**

Plan : Search for more information about Melanie C.

#E1 = Wikipedia[Melanie C]

Plan : Find out who featured on Melanie C's number one single Never Be The Same Again in 2000.

#E2 = LLM[Who featured on Melanie C's number one single Never Be The Same Again in 2000? Given context: #E1 ]

Plan : Search for more information about the featured artist.

#E3 = Wikipedia[ #E2 ]

Plan : Find out if the featured artist died in a car accident in 2002.

#E4 = LLM[Did #E2 die in a car accident in 2002? Given context: #E2 ]

**–SOLVER–**

Plan : Search for more information about Melanie C.

Evidence : Could not find [Melanie C]. Similar: ['Melanie C', 'Melanie C discography', 'Melanie C (album)', 'I Turn to You (Melanie C song)', 'I Want Candy', 'Northern Star (Melanie C album)', 'Mel B', 'Melanie', "I Don't Know How to Love Him", 'Reason (Melanie C album)']

Plan : Find out who featured on Melanie C's number one single Never Be The Same Again in 2000.

Evidence : Lisa Left Eye Lopes.

Plan : Search for more information about the featured artist.

Evidence : Lisa Nicole Lopes (May 27, 1971 – April 25, 2002), better known by her stage name Left Eye, was an American rapper and singer. She was a member of the R&B girl group TLC, alongside Tionne "T-Boz" Watkins and Rozonda "Chilli" Thomas. Besides rapping and singing backing vocals on TLC recordings, Lopes was the creative force behind the group, receiving more co-writing credits than the other members. She also designed some of their outfits and the stage for 'Fan Mail Tour' and contributed to the group's image, album titles, artworks, and music videos. Through her work with TLC, Lopes won four Grammy Awards.During her brief solo career, Lopes scored two US top 10 singles with "Not Tonight" and "U Know What's Up", as well as one UK number-one single with "Never Be the Same Again", the latter a collaboration with Melanie C of the British girl group Spice Girls. She also produced another girl group, Blaque, who scored a platinum album and two US top 10 hits. Lopes remains the only member of TLC to have released a solo album. On April 25, 2002, Lopes was killed in a car crash while shooting a documentary in Honduras. The vehicle she was in swerved off the road to avoid hitting another vehicle that stopped abruptly in front of them. The documentary was released as The Last Days of Left Eye and aired on VH1 in May 2007. Plan : Find out if the featured artist died in a car accident in 2002.

Evidence : Yes.

Lisa Left Eye Lopes .

## D.3  GSM8K

*ReAct*

Question: John decides to buy some birds. He got 50 dollars from each of his 4 grandparents. If each bird costs $20, how many wings did all the birds have?

Thought : I need to know how many birds John can buy with the money he got from his grandparents.

Action : Calculator[(50 * 4) / 20]

Observation : 10.0

Thought : Now I know how many birds John can buy. I need to know how many wings all the birds have. Action : Calculator[10.0 * 2]

Observation : 20.0

Thought : I now know the final answer

Action : Finish[ 20.0 wings ]

*ReWOO*

Question: John decides to buy some birds. He got 50 dollars from each of his 4 grandparents. If each bird costs $20, how many wings did all the birds have?

**–PLANNER–**

Plan : Calculate the total amount of money John received from his 4 grandparents.

#E1 = Calculator[50 * 4]

Plan : Calculate the total cost of all the birds.

#E2 = Calculator[20 * ( #E1 / 20)]

Plan : Calculate the total number of birds John can buy.

#E3 = Calculator[ #E1 / 20]

Plan : Calculate the total number of wings all the birds have.

#E4 = Calculator[ #E3 * 2]

**–SOLVER–**

Plan : Calculate the total amount of money John received from his 4 grandparents.

Evidence : 200

Plan : Calculate the total cost of all the birds.

Evidence : 200.0
Plan : Calculate the total number of birds John can buy.
Evidence : 10.0
Plan : Calculate the total number of wings all the birds have.
Evidence : 20.0
20

## D.4 STRATEGYQA

*ReAct*

Question: Answer with 'Yes' or 'No': Is it socially acceptable to wear an icon depicting crucifixion?

Thought : I'm not sure about the cultural and social norms regarding wearing a crucifixion icon.

Action : Google[Is it socially acceptable to wear a crucifixion icon?]

Observation : Protestants use the cross as the symbol of our faith, not a crucifix. ... Is it acceptable to wear a crucifix, if you are not practicing but still respect ...
Thought: The search results are not clear on whether it is socially acceptable or not. I need more information.

Action : LLM[What is the cultural and social significance of wearing a crucifixion icon?]

Observation : Symbol of faith.

Thought : The LLM's response is not helpful. I need to try a different approach.

Action : Wikipedia[Crucifixion icon cultural significance]

Observation : Could not find [Crucifixion icon cultural significance]. Similar: ['Crucifixion of Jesus', 'Christian symbolism', 'Cultural impact of Madonna', 'Jesus in Christianity', 'Jesus', 'Depiction of Jesus', 'Ascension of Jesus', 'Transfiguration of Jesus', 'Repatriation (cultural property)', 'Limbo']
Thought: I need to rephrase my search query to get more relevant results.

Action : Google[Cultural significance of wearing a crucifixion icon]

Observation : A defining symbol of the Christian faith, the cross is recognized globally as a sign of love, loss, salvation and gratitude, and is depicted in ...
Thought: Based on the search results, it seems that wearing a crucifixion icon is socially acceptable in Christian cultures.

Action : LLM[What are the cultural norms regarding wearing a crucifixion icon?]

Observation : Varies.

Thought : The LLM's response is not helpful. I need to make a decision based on the information I have gathered.

Action : Finish[ It is generally socially acceptable to wear a crucifixion icon in Christian cultures. ]

*ReWOO*

Question: Answer with 'Yes' or 'No': Is it socially acceptable to wear an icon depicting crucifixion?

**–PLANNER–**
Plan : Search for more information about crucifixion
#E1 = Wikipedia[Crucifixion]
Plan : Find out if it is socially acceptable to wear an icon depicting crucifixion
#E2 = LLM[Is it socially acceptable to wear an icon depicting crucifixion? Given context: #E1 ]
**–SOLVER–**
Plan : Search for more information about crucifixion

Evidence : Crucifixion is a method of capital punishment in which the victim is tied or nailed to a large wooden cross or beam and left to hang until eventual death. It was used as a punishment by the Persians, Carthaginians and Romans, among others. Crucifixion has been used in parts of the world as recently as the 21st century.The crucifixion of Jesus is central to Christianity, and the cross (sometimes depicting Jesus nailed to it) is the main religious symbol for many Christian churches.

Plan : Find out if it is socially acceptable to wear an icon depicting crucifixion

Evidence : Yes.

Yes.

## D.5 PHYSICSQUESTION

*ReAct*

Question: During their physics field trip to the amusement park, Tyler and Maria took a rider on the Whirligig. The Whirligig ride consists of long swings which spin in a circle at relatively high speeds. As part of their lab, Tyler and Maria estimate that the riders travel through a circle with a radius of 5.7 m and make one turn every 7.1 seconds. Determine the speed of the riders on the Whirligig.

Thought : I need to calculate the speed of the riders on the Whirligig.

Action : WolframAlpha[speed of a rider on a whirligig with a radius of 5.7 m and one turn every 7.1 seconds]

Observation : Wolfram Alpha wasn't able to answer it

Thought : I need to calculate the speed using the formula for angular velocity.

Action : Calculator[(2 * 3.14 * 5.7) / 7.1]

Observation : 5.041690140845071

Thought : I now know the final answer

Action : Finish[ The riders on the Whirligig travel at a speed of 5.041690140845071 m/s. ]

*ReWOO*

---

Question: During their physics field trip to the amusement park, Tyler and Maria took a rider on the Whirligig. The Whirligig ride consists of long swings which spin in a circle at relatively high speeds. As part of their lab, Tyler and Maria estimate that the riders travel through a circle with a radius of 5.7 m and make one turn every 7.1 seconds. Determine the speed of the riders on the Whirligig.

---

**–PLANNER–**

`Plan` : Use the formula for linear velocity to calculate the speed of the riders.

`#E1` = LLM[What is the formula for linear velocity?]

`Plan` : Identify the values needed to solve for the speed of the riders.

`#E2` = LLM[What values are needed to solve for linear velocity in this problem?]

`Plan` : Use the given information to calculate the speed of the riders.

`#E3` = Calculator[2 * 3.14 * 5.7 / 7.1]

---

**–SOLVER–**

`Plan` : Use the formula for linear velocity to calculate the speed of the riders.

`Evidence` : v = d/t

`Plan` : Identify the values needed to solve for the speed of the riders.

`Evidence` : Time, distance.

`Plan` : Use the given information to calculate the speed of the riders.

`Evidence` : 5.041690140845071

`5.041690140845071 m/s`

---

## D.6 SPORTS UNDERSTANDING

*ReAct*

---

Question: Determine whether the following statement or statements are plausible or implausible: Jorge Soler entered the attacking zone

---

`Thought` : I need to determine if Jorge Soler entered the attacking zone.

`Action` : LLM[Did Jorge Soler enter the attacking zone?]

`Observation` : Yes.

`Thought` : I now know the final answer.

`Action` : Finish[ `Plausible` ](*wrong*)

---

*ReWOO*

---

Question: Determine whether the following statement or statements are plausible or implausible: Jorge Soler entered the attacking zone

---

**–PLANNER–**

`Plan` : Use Google to search for news articles about Jorge Soler entering the attacking zone.

`#E1` = Google[Jorge Soler attacking zone]

---

**–SOLVER–**

`Plan` : Use Google to search for news articles about Jorge Soler entering the attacking zone.

`Evidence` : Jorge Soler career batting statistics for Major League, Minor League, and postseason baseball.

`Implausible`

---

## D.7 SOTUQA

*ReAct*

Question: Based on State of the Union Address 2023: What are the difference in the roles of McConnell and Chuck Schumer ?

`Thought` : I need to find information about the roles of McConnell and Chuck Schumer in the State of the Union Address 2023.

`Action` : SearchSOTU[McConnell and Chuck Schumer roles in State of the Union Address 2023]

`Observation` : Following is a transcript of President Biden's State of the Union address in 2023.
Mr. Speaker. Madam Vice President. Our First Lady and Second Gentleman.
Members of Congress and the Cabinet. Leaders of our military.
Mr. Chief Justice, Associate Justices, and retired Justices of the Supreme Court.
And you, my fellow Americans.
I start tonight by congratulating the members of the 118th Congress and the new Speaker of the House, Kevin McCarthy.
Mr. Speaker, I look forward to working together.
I also want to congratulate the new leader of the House Democrats and the first Black House Minority Leader in history, Hakeem Jeffries.
Congratulations to the longest serving Senate Leader in history, Mitch McConnell.
And congratulations to Chuck Schumer for another term as Senate Majority Leader, this time with an even bigger majority.
And I want to give special recognition to someone who I think will be considered the greatest Speaker in the history of this country, Nancy Pelosi.

`Thought` : The search result did not provide the information I was looking for. I need to try a different approach.

`Action` : LLM[What are the differences in the roles of Mitch McConnell and Chuck Schumer in the State of the Union Address 2023?]

`Observation` : McConnell: Preside. Schumer: Deliver response.

`Thought` :I now know the roles of McConnell and Chuck Schumer in the State of the Union Address 2023.

`Action` : Finish[ McConnell presided over the address while Chuck Schumer delivered the response. ]

*ReWOO*

Question: Based on State of the Union Address 2023: What are the difference in the roles of McConnell and Chuck Schumer ?

**–PLANNER–**
`Plan` : Use SearchSOTU to find relevant sections of the State of the Union Address 2023 that mention the roles of McConnell and Chuck Schumer.
`#E1` = SearchSOTU["McConnell Chuck Schumer roles difference"]
`Plan` : Summarize the differences in the roles of McConnell and Chuck Schumer.
`#E2` = LLM["Summarize the differences in the roles of McConnell and Chuck Schumer based on `#E1` ."]
**–SOLVER–**
`Plan` : Use SearchSOTU to find relevant sections of the State of the Union Address 2023 that mention the roles of McConnell and Chuck Schumer.

Evidence : Following is a transcript of President Biden's State of the Union address in 2023.
Mr. Speaker. Madam Vice President. Our First Lady and Second Gentleman.
Members of Congress and the Cabinet. Leaders of our military.
Mr. Chief Justice, Associate Justices, and retired Justices of the Supreme Court.
And you, my fellow Americans.
I start tonight by congratulating the members of the 118th Congress and the new Speaker of the House, Kevin McCarthy.
Mr. Speaker, I look forward to working together. I also want to congratulate the new leader of the House Democrats and the first Black House Minority Leader in history, Hakeem Jeffries.
Congratulations to the longest serving Senate Leader in history, Mitch McConnell.
And congratulations to Chuck Schumer for another term as Senate Majority Leader, this time with an even bigger majority.
And I want to give special recognition to someone who I think will be considered the greatest Speaker in the history of this country, Nancy Pelosi.
Plan : Summarize the differences in the roles of McConnell and Chuck Schumer.
Evidence : McConnell is Senate Majority Leader, Schumer is Senate Minority Leader.
McConnell is Senate Majority Leader, Schumer is Senate Minority Leader.

## D.8 OTHER OPEN-DOMAIN CURATED TASKS

Following are several curated tasks featuring real-world ALM use cases. We additionally provide a wider range of retrieval and acting tools to demonstrate the potential of `ReWOO`.

Additional tools include:

*Yelp[input]*: Worker that gives restaurant information including reviews, ratings and prices from Yelp. Input should be a search query.

*Twitter[input]*: Worker that searches results from Twitter. Useful when you need to find tweets about a topic. Input should be a search query.

*Location[input]*: Worker that retrieves user's current location. Input should be empty

*Time[input]*: Worker that retrieves current time. Input should be empty

*Email[input]*: Worker that can send Emails. Useful when you need to send someone email. Input should be in three parts: the target email address, subject and body, separated by a semicolon.

*Stock[input]*: Worker that retrieves current stock market analysis and recommendations. Input should be empty.

*TradeStock[input]*: Worker that connects to BackTrader to operate a trading strategy. Input should be in two parts, Stock ticker and indicator level, separated by semicolon.

*Draw[input]*: Worker that can draw and save a picture based on your prompt. Input should be a descriptive prompt for your picture.

---

Trying to get some coffee. Is there a recommended Cafe nearby?

**–PLANNER–**
Plan : Find out the user's current location. #E1 = Location[]
Plan : Search for recommended Cafe near the user's current location.
#E2 = Yelp[Cafe near #E1 ]

**–SOLVER–**
Plan : Find out the user's current location.
Evidence : Jersey City, NJ, 07302
Plan : Search for recommended Cafe near the user's current location.

Evidence : Semicolon Cafe. Ratings: 4.5 stars; Price range is $$. Phone number: (201)332-8148. Reviewers say: Semicolon Cafe is one of my favorite go tos in Jersey City! I love the flavors in their sandwiches and their lattes are wonderful. The staff are also super nice and make you feel welcomed. Their shaved ice is tasty, sweet, and the perfect treat after a long day. This place is a 10/10, would recommend.

Yes, Semicolon Cafe is 1.6 miles away and has 4.5 stars.

---

Write prof Yann LeCun an email, asking if he's hiring TA for 2023 Fall DS-1008 Deep Learning.

**–PLANNER–**

Plan : Retrieve user's current location.

#E1 = Location[]

Plan : Retrieve current time.

#E2 = Time[]

Plan : Search for Yann LeCun's email address.

#E3 = Google[Yann LeCun's email address]

Plan : Compose email with target address, subject and body.

#E4 = Email[ #E3 ; DS-1008 Deep Learning TA 2023 Fall; Hi Prof. Yann LeCun, I am writing to inquire about any TA position openings for your DS-1008 Deep Learning course in 2023 Fall semester. I am very interested in the course and would love to contribute to the class. Thank you for your time. Best regards, [Name] ]

**–SOLVER–**

Plan : Retrieve user's current location.

Evidence : Jersey City, NJ, 07302

Plan : Retrieve current time.

Evidence : 6:12 PM EST, May 8, 2023

Plan : Search for Yann LeCun's email address.

Evidence : @lecun.org. @fb.com. @courant.nyu.edu.

Plan : Compose email with target address, subject and body.

Evidence : email sent

Subject: Request for TA Position for 2023 Fall DS-1008 Deep Learning
Dear Professor LeCun,
I am writing to inquire if you are hiring teaching assistants for your 2023 Fall DS-1008 Deep Learning course. I am currently located in Jersey City, NJ, 07302 and the time here is 5:44 PM EST, May 4, 2023.
I am passionate about deep learning and I believe my knowledge and experience in the field would make me an ideal candidate for this position. I am eager to learn more about the position and discuss how I can contribute to the course.
Thank you for your time and consideration.
Sincerely, [Your Name]

---

Is there any stock you recommend buying today?

**–PLANNER–**

Plan : Retrieve current location.

#E1 = Location[]

Plan : Retrieve current time.

#E2 = Time[]

Plan : Retrieve stock market analysis and recommendation.

#E3 = Stock[]

Plan : Connect to BackTrader to operate a trading strategy.

#E4 = TradeStock[ #E3 ; indicator level]

**–SOLVER–**

Plan : Retrieve current location.

Evidence : Jersey City, NJ, 07302

Plan : Retrieve current time.

Evidence : 5:44 PM EST, May 4, 2023

Plan : Retrieve stock market analysis and recommendation.

Evidence : DPST shows BUY_AND_HOLD signal at $4.03. Confidence: 7.0

Plan : Connect to BackTrader to operate a trading strategy.

Evidence : No evidence found

Yes, DPST at $4.03 with a BUY_AND_HOLD signal and a confidence of 7.0.

---

Draw a logo for my project – It modularizes Planner, Worker, and Solver to solve hard tasks like humans. It represents the embryo of AGI in an efficient and scalable way.

**–PLANNER–**

Plan : Generate a sketch for the logo.

#E1 = Draw[A sketch of a Planner, a Solver, and bunch of Workers connected together, representing the embryo of future AGI in a cost efficient way.]

Plan : Enhance the sketch to make it look more professional.

#E2 = Draw[Enhance #E1 to make it look more professional.]

**–SOLVER–**

Draw a logo for my project – It modularizes Planner, Worker, and Solver to solve hard tasks like humans. It represents the embryo of AGI in an efficient and scalable way.

Plan : Generate a sketch for the logo.

Evidence : Drawing saved to my_pic.png

Plan : Enhance the sketch to make it look more professional.

Evidence : No evidence found

Drawing saved to my_pic.png. Enhance the sketch to make it look more professional.

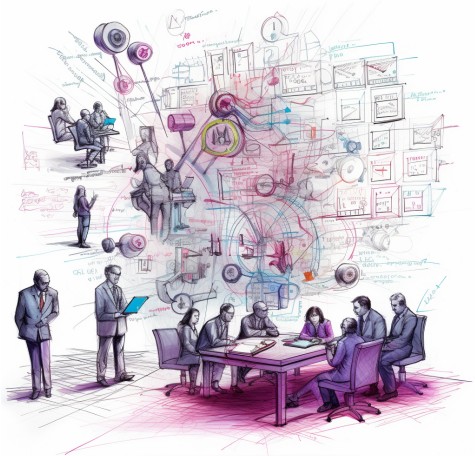

