# OpenReview forum: "DECOUPLING REASONING FROM OBSERVATIONS FOR EFFICIENT AUGMENTED LANGUAGE MODELS"
_ICLR.cc/2024/Conference — Submitted to ICLR 2024_

### Official Review · Reviewer_Ewxu · 2023-10-30

**Soundness:** 3 good
**Presentation:** 3 good
**Contribution:** 2 fair
**Rating:** 5
**Confidence:** 4

**Summary:**

This paper proposed a novel modular-based prompting paradigm for Augmented Language Models called ReWOO (Reasoning WithOut Observation), which decouples the reasoning process from external observations. As a result, ReWOO largely reduced the token consumption, which is a major limiting factor to LLMs' applications. Comprehensive experiments on several multi-step reasoning tasks show the effectiveness of the proposed framework. Besides, by offloading reasoning from LLMs to smaller language models, end-to-end training is possible.

**Strengths:**

* The novel modular framework ReWOO proposed in this paper is one of the very first works showing concern about the token efficiency given LLMs under current research trends.

*. The paper has a good structure for presenting the methodology and experimental results.

**Weaknesses:**

1. ReWOO requires predefined action space as shown in Section 3.1 SETUPS. Although it supports three-fold tasks and several reasoning capabilities, ReWOO is not easy to extend. Obviously, more annotations and re-training/tuning are non-trivial if more tools are required.

2. The experiment settings are not clear enough in the submission version. For Direct Prompt and CoT, which LLMs are employed? Do they also leverage GPT-3.5-turbo? It will be important to make a fair comparison between these distinct experimental settings.

3. Since ReWOO is a modular framework, for the Planner, the quality and diversity of the exemplars can influence performance. The paper lacks experiments for evaluating different exemplars and the proposed method's robustness under such scenarios.

4. The reason why the Direct and CoT could be better than ALMs is not clear. From my understanding, CoT, for example, generates all human-like thoughts in the decoding process, which is different from ReWOO encoding all the plans on the encoding side. One of the reasons for the performance drop is due to tool failure. However, is there any other reason, e.g. the architecture? Detailed discussions need to be covered.

5. What is the relationship between this paper and the question decomposition literature, especially in the QA area?

**Questions:**

1. For the experiment shown in Figure 5, the original intention is to show adding extraneous tools could degrade the model performance. But why is HotpotQA selected for this experiment? For HotpotQA, ReWOO achieves the overall best performance than CoT. Is it more suitable to choose TriviaQA or GSM8K for this analysis? Besides, the exemplars for HotpotQA is 6, which is larger than others. Will this be another influence for conducting such an analysis?

2. The error or performance degradation may come from two aspects: one is tool failure as stated in this paper, and the other is planner failure, from my point of view. Because the planner is not perfect, it could produce unintended consequences/plans that are not aligned with the tasks or datasets. Is there any analysis on this possible cause of failures?

---

### Official Review · Reviewer_s2Wj · 2023-11-02

**Soundness:** 3 good
**Presentation:** 3 good
**Contribution:** 3 good
**Rating:** 6
**Confidence:** 3

**Summary:**

This paper introduces ReWOO (Reasoning WithOut Observation) to separate the reasoning process from external observations in tool invocation, which significantly decreases computational complexity. Moreover, ReWOO also demonstrates stronger performance than ReACT and robustness under tool failure scenarios. The authors also employ instruction fine-tuning in smaller language models to reduce model parameters. The paper is well-organized, and the method proves to be useful.

**Strengths:**

1. The paper is well-organized and easy to read with clear explanations of the technical details.
2. The method of reasoning without observation decreases computational complexity significantly compared to ReACT, which is useful for invoking tools in large language models.
3. The experiments conducted in seven NLP benchmarks demonstrate the effectiveness of the method.

**Weaknesses:**

1. The planner aims to compose a solution blueprint, which plays an important role in the ReWOO method. However, the author does not introduce a way to validate the blueprint provided by the planner.
2. In certain situations, the model needs to determine the next action based on previous observations. In my opinion, the method seems challenging to implement.
3. The NLP benchmarks contain fewer steps (less than 6, with most having fewer than 4 steps). How does ReWOO perform in more complex scenarios?

**Questions:**

See above.

---

### Official Review · Reviewer_1Xgs · 2023-11-03

**Soundness:** 2 fair
**Presentation:** 2 fair
**Contribution:** 2 fair
**Rating:** 3
**Confidence:** 3

**Summary:**

The paper studies cost-efficiently augmenting LLMs with external tools, and propose a prompting technique, called ReWOO. ReWOO solves reasoning problems in three steps with three modular components. First, a Planner generates a plan to solve the task using the LLM's foreseeable reasoning ability. The Worker retrieves evidence from tools. The Solver synthesizes the plan and evidence into a final solution. Using this way, ReWoo can streamline LLM generation without putting LLMs on halt, which reduces the cost of stateless API service. Experiments over 6 NLP benchmarks show ReWOO uses fewer tokens than ReAct and achieves higher accuracy on average. The paper also explores the possibility of distilling the planning ability to smaller LLM.

**Strengths:**

The paper presents experiments on several datasets covering multiple tasks.

The results suggest it can effectively cut down the cost compared to ReACT if using stateless APIs.

The paper is well written and easy to understand.

**Weaknesses:**

1. The comparison needs to be fairer. The paper emphasizes the cost. In Table 2, CoT uses significantly less cost than ReWOO, a more proper comparison would need using CoT with self-consistency decoding to balance the cost

2. Following 1, at the same time, the performance of ReWoo does not show consistent gain compared to CoT and even lags CoT for some datasets when CoT uses much less compute

3. The planning, working, and solving procedure is similar to lots of existing work (e.g., [1][2][3] some of them are missing references). The paper mainly differs in that the workers can be tools.

4. The paper focuses on a somewhat narrow problem. Mainly for reducing the cost for using stateless APIs. Providers of APIs can build APPS that more closely integrate the tools.

[1] Plan-and-Solve Prompting: Improving Zero-Shot Chain-of-Thought Reasoning by Large Language Models. Want et al., ACL 2023

[2] Decomposed Prompting: A Modular Approach for Solving Complex Tasks. Khot et al., ICLR 2023

[3] BREAK It Down: A Question Understanding Benchmark. Wolfson et al., TACL 2020

**Questions:**

See weakness.

---

### Official Review · Reviewer_EFq2 · 2023-11-04

**Soundness:** 2 fair
**Presentation:** 3 good
**Contribution:** 2 fair
**Rating:** 3
**Confidence:** 5

**Summary:**

Motivated by the fact that previous reasoning and planning methods such as ReAct use repeated tokens in a language model that results in excessive cost, this paper proposes ReWOO (Reasoning without observation), which separates the planning and the tool execution steps. Evaluated on common knowledge reasoning tasks (including HotpotQA TriviaQA, SportsUnderstanding, StrategyQA), Arithmetic and Scientific Reasoning (including GSM8K, PhysicsQuestions), and Curated (including SOTUQA, recommendation for restaurants, and others), results show that the propose method outperform ReAct while significantly reduces the inference cost. Moreover, the authors augmented 2000 planner instruction data to finetune an Alpaca model and showed that ReWOO method can perform similarly to a GPT 3.5 model.

**Strengths:**

1. The paper is motivated by an important problem in reasoning at inference time. Although many methods (such as ReAct) are effective, indeed they introduced more computational costs. ReWOO can significantly reduce such repetition and correspondingly the cost, while maintaining the performance.
2. Results show that ReWOO can be applied to smaller models with distillation, which further makes it more efficient.

**Weaknesses:**

1. Despite the efficiency of the proposed method,  although the authors claim that "This paradigm enables ReWOO to tackle multi-step and complex tasks, particularly those where a subsequent step depends on the observations of prior steps", it does not seem that ReWOO is actually able to tackle multi-step complex tasks where subsequent step (of planning and reasoning) depends on the observations of prior steps. This would greatly limit the use case of the propose method. This is the inevitable bottleneck of separating planning and observation. Furthermore, for many multihop qa questions, google search or other retrieval tools can actually return results from one tool execution, which may not require methods such as ReAct to execute multiple reasoning and planning steps once the observation satisfies the performance; in comparison, ReWOO would always execute all planned executions. It would not be able to backtrack (i.e., issue new tool arguments) based on observations.
2. The huge gap between ReWOO/ReAct and Direct and CoT on triviaQA and GSM8K makes the evaluation questionable. I understand that sometimes tool-call is not necessary to get an answer correct, but these two tasks are either knowledge-intensive or could greatly benefit from using a calculator. The much lower results for ReAct and ReWOO suggests that either the tool or the prompt is not optimized compared to the direct method. If the authors hypothesize that the issue is due to tool selection, further investigation is required. For example, what the results would be if only the optimal tool is employed. Otherwise, the benefit of using ReWOO is not convincing.
3. Similarly, there is not enough information about the curated datasets (either in the appendix or in the project repo). This prevents detailed understanding of the performance on these tasks.

**Questions:**

1. Would different number of exemplars (n) make any differences for Table 2?
2. What happens when intermediate tools fail? i.e., the model will incorporate hallucinated information? Why would this be a major win of ReWOO compared to ReAct?
3.  What are the differences in these tools (e.g., google search vs. wikipedia, Calc vs. WolAlf)? How do you choose which tools to provide for each task? When would the LLM tool be useful?

---

### Meta-Review · Area_Chair_QhnY · 2023-11-30

**Metareview:**

The reviewers raised questions regarding the multi-step complex tasks, the methods of evaluation and comparison, and the validation of the blueprint provided by the planner. However, the authors did not provide a rebuttal. Considering the current ratings of 6/5/3/3, I recommend rejecting the submission.

**Justification For Why Not Higher Score:**

See metareview

**Justification For Why Not Lower Score:**

N/A

---

### Decision · Program_Chairs · 2024-01-16

Reject